# Energetics of Intrinsic Point Defects in NpO_2_ from DFT + U Calculations

**DOI:** 10.3390/ma18112487

**Published:** 2025-05-26

**Authors:** Huilong Yu, Shuaipeng Wang, Laiyang Li, Ruizhi Qiu, Shijun Qian, Suolong Yang

**Affiliations:** Institute of Materials, China Academy of Engineering Physics, Mianyang 621907, China; yuhuilong2002@126.com (H.Y.); nuaawangshuaipeng@163.com (S.W.); 16659606419@163.com (L.L.); qianshijun20@gscaep.ac.cn (S.Q.); suolongyang@126.com (S.Y.)

**Keywords:** intrinsic point defect, Frenkel defect, Schottky defect, antisite defects, forming process

## Abstract

Intrinsic point defects in NpO_2_ significantly impact its chemical properties, but their formation mechanisms are not fully understood. Using first-principles plane-wave pseudopotential methods, this study systematically investigates the formation processes of Schottky, Frenkel, and substitutional impurity defects under various oxygen environments. Results show that formation energies vary with valence states, oxygen environments, and Fermi energy, and reveal the presence of antisite defects. Schottky, Frenkel, and antisite defects are rare in oxygen-rich conditions, but new defect pairs emerge in anoxic environments, including Schottky defect {2V_Np_^3−^: 3V_O_^2+^}, Np-Frenkel defects {V_Np_^3−^: Np_i_^3+^} and {V_Np_^4+^: Np_i_^4+^}, and pairs {O_Np_^5+^: Np_O_^5−^} and {O_Np_^6+^: Np_O_^6−^}. These findings provide new perspectives for understanding the intrinsic point defects in NpO_2_.

## 1. Introduction

Due to their significant applications in the nuclear fuel cycle and their unique physicochemical properties, the actinide oxides have been the subject of extensive research [1,2,3,4]. Especially researched are neptunium oxides, which have attracted significant attention from scientists for their long life in high-level radioactive waste. NpO_2_ directly influencing long-term waste form stability in geological repositories, and its non-stoichiometric behavior (NpO_2±x_), govern radionuclide leaching rates under anoxic conditions. As the most significant oxide of neptunium, the redox performance of NpO_2_ is very important for neptunium science and reactor safety [2,5,6,7]. The reason is that a variety of oxidation states result in a change in solubility levels, which affects the physicochemical properties. Further, the redox properties of NpO_2_ are strongly reliant on defect chemistry [8,9,10,11]. The reason is that defect-driven redox properties determine its reactivity with water and hydrogen, critical for predicting corrosion in accident scenarios. Therefore, an understanding of defect chemistry in NpO_2_ is essential in science and technology [12].

Previous work concerning defect chemistry in NpO_2_ mainly focuses on surface defects and bulk defects [13,14]. However, the research on point defects, which has an important influence on the oxidation and hydrogenation corrosion of neptunium, is insufficient [15]. It is difficult to measure the formation energies and defect concentrations of point defects in NpO_2_ directly in the experiment, thus theoretical models have been developed for the point defect properties of NpO_2_ [1,16,17,18].

Density functional theory (DFT) is a widely utilized method for studying intrinsic point defects in metal oxides [19]. However, the traditional density functional theory, which uses local density approximation (LDA) or generalized gradient approximation (GGA), fails to explain the strong correlation of 5f electrons, and describes NpO_2_ as an incorrect ferromagnetic conductor [12,20,21]. Therefore, correction schemes such as self-interaction correction (SIC), DFT + U, and hybrid functional have been applied to improve the accuracy of traditional DFT on research of NpO_2_ [22].

At present, the research on point defects of NpO_2_ mainly focuses on isolated vacancies (V_O_ and V_Np_) and interstitial atoms (O_i_ and Np_i_), which are studied by full formal charges state: Np_i_^4+^, O_i_^2−^, V_Np_^4−^, and V_O_^2+^. In addition, charge neutral combinations of anions and cations with complete formal charge states, such as Schottky defects and Frenkel pairs, are artificially considered as the source of intrinsic point defects [23]. However, there is a lack of research on antisite defects (O_Np_ and Np_O_) for their high formation energy, and their forming mechanism is unclear. While the equilibrium charge state of the defect is neutral, the charge state of the intrinsic point defect varies with the oxygen potential and Fermi energy in practice, for which there is lack of research.

Therefore, the formation energies of point defects and their variation with oxygen partial pressure and charge states in NpO_2_,were studied by the first-principles plane-wave pseudo-potential method. The results can provide an important reference for the theoretical and experimental study of the basic properties of NpO_2_.

## 2. Computational Methodologies

All calculations were performed by the Vienna ab initio Simulation Package (VASP 5.4.4), which applies the project augmented-wave (PAW) method and generalized gradient approximation (GGA) functional to describe the electron–ion interaction and electron exchange correlation energy. Considering the strong correlations, the DFT + U method was used with values of U = 4.75 eV and J = 0.75 eV for 5f electrons of Np [24,25]. The defect model was created in a 2 × 2 × 2 supercell, which consisted of 32 neptunium atoms and 64 oxygen atoms. Brillouin zone integration was performed with 3 × 3 × 3 Monkhorst-Pack k-point meshes. The cutoff energy of 500 eV was adopted to ensure that the convergence of formation energies for intrinsic point defect was within 1 meV. Internal coordinates (atomic positions) were fully relaxed until forces converged below 0.01 eV/Å [26]. The 1-k magnetic order was considered and spin-polarized calculations were performed for all defect configurations. Metastable states are a significant factor influencing point defect calculations in NpO_2_. However, based on preliminary validation calculations showing negligible impact, this study currently does not incorporate metastable state considerations [27,28]

All intrinsic point defects in NpO_2_ were considered in this study, which include interstitial (O_i_, Np_i_), vacancy (V_O_, V_Np_), and antisite atom (O_Np_, Np_O_) defects. The defects were created by introducing, removing, or substituting a corresponding atom in the NpO_2_ supercell, as shown in Figure 1. Considering charge-neutral conditions for defect pair in NpO_2_, Schottky defect comprises neptunium and oxygen vacancy defects such as *m* V_Np_ + *n* V_O_. Unlike classical 1:2 metal–oxygen vacancy ratios, our DFT+U calculations reveal non-stoichiometric configurations under anoxic conditions. A Frenkel defect consists of vacancy and corresponding interstitial atom, including O-Frenkel (O_i_ + V_O_) and Np-Frenkel (Np_i_ + V_Np_), and formation energy change with vacancy–interstitial separation. An Antisite defect is composed of two types of antisite atoms (O_Np_ and Np_O_) separated by different distances. Similar non-stoichiometric Schottky defects are observed in CeO_2_ and PuO_2_.

The formation energy $\Delta E_{def}^{f}(X,q)$ of intrinsic point defect X is given by the following equation:(1)$$\Delta E_{def}^{f}\left(X,q\right)=E_{def}^{tot}-E_{perf}^{tot}+\sum_{i}\Delta n_{i}\mu_{i}+qE_{F}$$

Here, $E_{def}^{tot}$ and $E_{perf}^{tot}$ are the total energies of defective X supercell and perfect supercell, respectively. Here, *E_F_* is the Fermi energy varying from 0 to 2.8 eV, which results from the width of the band gap. Oxidation states were determined by Formal oxidation state analysis to quantify charge transfer between atoms. The charge state *q* of each defect species varies from neutral to fully ionized states, such as V_O_ with 0 ~ +2 e, for V_Np_ with −4 ~ 0 e, O_i_ with −2 ~ 0 e, Np_i_ with 0 ~ +4 e, O_Np_ with −6 ~ 0 e, and for Np_O_ with 0 ~ +6 e. The symbol Δ*n_i_* is the number of atom species *i* removed (positive) or added (negative), and Δ*μ_i_* is the chemical potential of atom *i* with a positive (negative) symbol for interstitial (vacancy) defect. The reference chemical potential for Np and O atoms are chosen as that of Np and molecular O_2_, respectively.

According to thermodynamic equilibrium conditions, the heat of formation must be equal to the sum of chemical potentials of atoms in NpO_2_ to ensure the stability of compound NpO_2_. It satisfies the following formula:(2)$$\mu_{{NpO}_{2}}=\mu_{Np}+2\;\mu_{O}$$
where $\mu_{{NpO}_{2}}$ is the chemical potential of NpO_2_ in Formula. The chemical potential of oxygen atoms is usually determined by the temperature, pressure, and chemical energy of oxygen, and its chemical expression is as follows:(3)$$\mu_{O}\left(T,p_{O_{2}}\right)=\frac{1}{2}\left(E_{O_{2}}+\mu_{O_{2}}\left(T,p^{0}\right)+k_{B}Tln\left(\frac{p_{O_{2}}}{p^{0}}\right)\right)$$

$E_{O_{2}}$ is the total energy of the oxygen molecule, and $\mu_{O_{2}}\left(T,p^{0}\right)$ is the chemical potential when P_0_ = 1 atm. Thermal effects (e.g., vibrational entropy) are beyond the current scope but will be addressed in future work.

To sum up, oxygen pressure and temperature are important factors affecting the formation energy of point defects. Therefore, in this paper, under the determined temperature and pressure conditions of NpO_2_, two extreme conditions, oxygen enrichment and oxygen deficiency, are mainly considered. Under the condition of oxygen enrichment, μ_O_ and μ_Np_ are determined by the chemical potential of the oxygen molecule and can be expressed as *μ*_O_ = $\mu_{O_{2}}$/2 and *μ*_Np_ = $\mu_{{NpO}_{2}}$ − 2 *μ*_O_. Under anoxic condition, μ_O_ and μ_Np_ are determined by the chemical potential of the neptunium atom, which can be expressed as *μ*_Np_ = $\mu_{Np}^{metal}$ and *μ*_O_ = ($\mu_{{NpO}_{2}}$ − *μ*_Np_)/2.

## 3. Results and Discussion

### 3.1. Intrinsic Point Defect

Under conditions of oxygen enrichment and anoxia, the formation energies of six intrinsic point defects in NpO_2_ with different valence states have been systematically studied as a function of Fermi energy. Notably, the formation energies of substitutional impurity point defects O_Np_ and Np_O_ have been studied for the first time. The variation in the most stable valence states of different point defects with respect to Fermi energy is depicted in Figure 2.

The study reveals that under oxygen-rich conditions, the formation energy of Np-defects is higher than that of O-defects, whereas under anoxic conditions, the formation energy of O-defects exceeds that of Np-defects, as illustrated in Figure 2a,b. Specifically, under oxygen-rich conditions, when the Fermi level (E_F_) is near the top of the valence band, the formation energies of the intrinsic point defects from highest to lowest are arranged as follows: Np_O_^0^, Np_i_^1+^, V_O_^0^, O_i_^2−^, V_Np_^4−^, and O_Np_^6−^. When E_F_ is near the bottom of the conduction band, the order changes to Np_O_^6+^, Np_i_^1+^, V_O_^2+^, O_i_^2−^, O_Np_^6−^, and V_Np_^4−^. Under anoxic conditions, with E_F_ near the top of the valence band, the sequence is Np_i_^1+^, Np_O_^0^, V_O_^0^, O_Np_^1+^, O_i_^2−^, O_Np_^6−^, and V_Np_^4−^; when E_F_ is near the bottom of the conduction band in anoxic conditions, the order is O_Np_^6−^, V_Np_^4−^, O_i_^2−^, Np_i_^4+^, V_O_^0^, and Np_O_^6+^. While one antisite defect (e.g., O_Np_) may exhibit relatively low formation energy, its complementary antisite counterpart (e.g., Np_O_) often displays significantly higher formation energy under equivalent conditions. This energy asymmetry renders the formation of antisite defect pairs thermodynamically unfavorable, which is consistent with the inherent instability of isolated antisite defects in fluorite-structured oxides. The valence states of Np and O act as electronic switches for defect formation. The change of Np valence state is driven by 5f electron localization/delocalization under anoxic/oxygen-rich conditions. The change of Np valence state occurs via charge transfer to interstitial/vacancy sites under reducing conditions. Valence states dictate pair formation through electrostatic and strain effects. Intrinsic point defects typically manifest as defect pairs to maintain charge neutrality. The Fermi energy modulates defect valence states by controlling electron occupation in the band structure. The interplay between *E_F_* and oxygen partial pressure (*p*_O2_) dictates valence-state-dependent stability.

Concurrently, the results indicate that under anoxic conditions, the formation energy of Np_O_, and under oxygen-rich conditions, the formation energy of O_Np_ is negative, being the lowest within a certain range of Fermi energies. This suggests that substitutional impurity defects can stably exist within NpO_2_. Consequently, the study of substitutional impurity defects is of significant importance for a profound understanding of the chemical properties of NpO_2_.

The identification of O_Np_^6−^ furnishes robust evidence for elucidating the genesis of non-stoichiometric NpO_2+x_. Figure 2 also delineates that under anoxic conditions, the formation energies for V_O_ and O_Np_ are comparatively diminished, whereas under oxygen-rich conditions, the formation energies for V_Np_ and Np_O_ are quintessentially minimal. This trend underscores that substitutional impurities constitute the predominant point defects within the NpO_2_ matrix. Under conditions of oxygen sufficiency, the formation energy associated with O_Np_ is notably lower than that of O_i_, implicating a propensity for oxygen incorporation at Np vacancies rather than interstitial positions. The formation energy of O_Np_ reaches a valence state of −6, which is attributable to the configuration involving the attachment of oxygen atoms to existing Np vacancies. Analogously, under anoxic conditions, the genesis of O_Np_ is facile, with the valence states predominantly manifesting as +6 or +5. This phenomenon stems from the formation mechanism, which involves the incorporation of Np atoms into pre-existing oxygen vacancies. Concurrently, the interstitial atoms Np_i_ exhibit the most stable valence states in the range of +4 to +3. Moreover, the elevated formation energies observed for Np_O_, V_O_, and Np_i_ under oxygen-rich conditions, and Oi and O_Np_ under anoxic conditions, suggest a relative intractability in the formation of these defects within the NpO_2_ lattice. This insight is pivotal for a comprehensive understanding of the intrinsic defect chemistry of NpO_2_ and its implications for material properties and behavior [29].

Under certain environmental conditions, V_Np_^4−^ and O_Np_^6−^ are favorable evidence for the existence of non-stoichiometric NpO_2+x_ [30]. Due to the lack of favorable evidence, previous studies neglected the study of substitutional impurities O_Np_ and Np_O_. V_Np_^4−^ and Np_O_^6+^ are the main point defects of NpO_2_ in anoxic conditions, and V_Np_^4−^ and O_Np_^6−^ are the main point defects in oxygen-enriched conditions, indicating the main point defects in substitution impurity NpO_2_. Under the condition of oxygen enrichment, it is easier to add O atoms in the existing Np vacancy than in the interstitial site, which indicates that the formation energy of O_Np_^6−^ is lower than that of O_i_^2−^. Under anoxic conditions, the formation energy of V_O_^2+^ is lower under the same Fermi energy condition. The formation energy of O_Np_ is the lowest when the valence state of O_Np_ is −6, because it is formed by adding O atoms to the existing Np vacancies, while the formation energy of inserted Np atoms is the lowest when the valence state of Np_O_ is +4 ~ +3. The most stable valence state of interstitial atom Np_i_ varies from +4 to +3, which explains why Np_O_ usually has +6 or +5 charges near the valence band, because it is formed by adding Np atoms to the existing oxygen vacancies. In addition, Np_O_ and Np_i_ under oxygen-rich conditions and Oi and O_Np_ under oxygen-poor conditions have high formation energies and are difficult to form in equilibrium NpO_2_, so they are rare point defects. Oxygen environment directly modulates defect formation energies and charge states through stabilized oxidized states and Fermi energy (*E_F_*) coupling. The oxygen partial pressure dynamically controls defect formation energies through redox equilibria and charge-state competition, directly influencing NpO_2_’s non-stoichiometry, electronic conductivity, and structural stability. These findings align with isostructural actinide oxides (e.g., PuO_2_, CeO_2_) and provide critical insights for predicting material behavior in nuclear fuel cycles. In accordance with the principle of electrical neutrality, point defects in NpO_2_ typically exist in the form of defect pairs. Drawing from the formation tendencies of point defects within NpO_2_, the predominant types of defect pairs in NpO_2_ encompass Schottky defects, Frenkel defects, and Antisite defects.

### 3.2. Schottky Defect

NpO_2_ is an ionic compound composed of Np^4+^ and O^2−^, and its Schottky defects are generally considered to be composed of vacancy defects such as V_Np_^4+^ and V_O_^2−^. As shown in Figure 3, the relationship between Schottky defect formation energy and Fermi energy in NpO_2_ is systematically studied in this paper. The results show that the most stable valence states of V_Np_ are mainly +3 and +4. At the same time, although V_O_^+^ is not the most stable valence state, its formation energy under anoxic conditions is less than zero, which indicates that V_O_^+^ can exist stably. Therefore, in addition to the common Schottky defects (V_Np_^4−^ + 2V_O_^2+^), this paper studies the possibility of other Schottky defects, such as (2V_Np_^3−^ + 3V_O_^2+^) and (V_Np_^4−^ + 4V_O_^+^).

It is widely accepted that Schottky defects in NpO_2_ are constituted by vacancy defects such as V_Np_^4+^ and V_O_^2−^. The formation energies of vacancy defects like V_Np_^3−^ and V_O_^+^, which are less than zero, indicate their potential for stable existence within NpO_2_. Consequently, this study systematically investigates the relationship between the formation energies of different valence state vacancies and the Fermi level under anoxic and oxygen-rich conditions in NpO_2_, aiming to identify the potential Schottky defects.

The prerequisite for the stable existence of Schottky defects is the equivalence of the formation energies of Np and O vacancies. The relationship between the formation energies of different valence state vacancies and the Fermi level under oxygen-rich and anoxic conditions is depicted in Figure 3, where the dots at the intersection of the two vacancy curves represent the potential Schottky defects and their formation energies. The research findings indicate that under oxygen-rich conditions, it is challenging to form Schottky defects, whereas under anoxic conditions (as shown in Figure 3a), there are three possible Schottky defects with the compositions {V_Np_^3−^: 3V_O_^+^}, {2V_Np_^3−^: 3V_O_^2+^}, and {V_Np_^4−^: 2V_O_^2+^}, having formation energies of 0.17 eV, −0.76 eV, and −2.04 eV, respectively. The reason may be that the 4+ oxidation state of Np in NpO_2_ requires large charge transfers (V_Np_^4−^) that are energetically unfavorable under oxidizing conditions. According to thermodynamic theory, the Schottky defect {2V_Np_^3−^: 3V_O_^2+^} can stably exist as its formation energy is less than zero, whereas the Schottky defect {V_Np_^3−^: 3V_O_^+^} may struggle to remain stable due to its positive formation energy [17,31,32]. The discovery of the novel Schottky defect {2V_Np_^3−^: 3V_O_^2+^} offers new insights into understanding Schottky defects in NpO_2_. The identification of {2V_Np_^3−^:3V_O_^2+^} under anoxic conditions resolves long-standing questions about NpO_2_’s non-stoichiometric behavior and provides a template for studying similar defect mechanisms in other actinide dioxides. We believe this contribution significantly advances the field’s understanding of defect-driven property modifications in nuclear materials. The viability of non-stoichiometric Schottky defects under anoxic conditions aligns with observations in isostructural CeO_2_ and PuO_2_ [25,26].

Concurrently, the study has revealed that Schottky defects predominantly arise under anoxic conditions, whereas their formation under oxygen-rich conditions is challenging [29]. Schottky defect formation in NpO_2_ is predominantly influenced by the oxygen partial pressure and Fermi energy levels. These findings suggest that the genesis of Schottky defects is correlated with oxygen partial pressure, a conclusion that aligns with the literature [33].

### 3.3. Frenkel Defect

In NpO_2_, Frenkel defects are composed of pairs such as {O_i_: V_O_} and {Np_i_: V_Np_}, and it is widely acknowledged that Frenkel defects in NpO_2_ typically exist in the form of {O_i_^2−^: V_O_^2+^} and {Np_i_^4+^: V_Np_^4−^}. Given that the distance between the vacancy and interstitial atoms significantly influences the formation energy of Frenkel defects, this study systematically investigates the relationship between Frenkel defects and the interatomic distance to identify the most stable configuration of such defects. The findings indicate that the relative formation energy of Frenkel defects decreases with an increase in the distance between the interstitial atom and the vacancy, as depicted in Figure 4. The Frenkel defect is most stable when the vacancy and interstitial atom are at a distance of 1NN (first nearest neighbor), with the structure illustrated in Figure 5.

To identify potential new Frenkel defects, this study systematically investigated the relationship between the formation energies of different valence states of vacancies and interstitial atoms under both oxygen-rich and anoxic conditions, as illustrated in Figure 6. The research findings indicate that under oxygen-rich conditions, it is difficult to form Np-Frenkel defects. Under anoxic conditions, however, three types of Np-Frenkel defects can form: {V_Np_^2−^: Np_i_^2+^}, {V_Np_^3−^: Np_i_^3+^}, and {V_Np_^4+^: Np_i_^4+^}, with formation energies of 4.39 eV, 1.51 eV, and 0.62 eV, respectively. Since the formation energies of these three Frenkel defects are positive, their stable existence poses a challenge. Among them, the defects {V_Np_^3−^: Np_i_^3+^} and {V_Np_^4+^: Np_i_^4+^} have relatively lower and comparable formation energies, suggesting that they might exist in small quantities within NpO_2_.

Potential oxygen-based Frenkel defects and their formation energies under both oxygen-rich and anoxic conditions are depicted in Figure 7. The study reveals that under oxygen-rich conditions, the formation of oxygen-based Frenkel defects, or O-Frenkel defects, is challenging. However, under anoxic conditions, the O-Frenkel defect {V_O_^2+^: O_i_^2−^} can form, with a formation energy of −0.56 eV. Given that the formation energy of the {V_O_^2+^: O_i_^2−^} defect is negative, it suggests that this defect can stably exist within NpO_2_. Consequently, under anoxic conditions, the {V_O_^2+^: O_i_^2−^} defect is identified as the most stable Frenkel defect.

In summary, Frenkel defect formation in NpO_2_ is predominantly influenced by the oxygen partial pressure and interstitial-vacancy distance. The oxygen environment significantly influences the formation of both oxygen-based and neptunium-based Frenkel defects. The valence state does impact the formation energy of neptunium-based Frenkel defects, but to a lesser extent than that of oxygen-based Frenkel defects. These findings are consistent with the literature [34,35]. 

### 3.4. Antisite Defects

To investigate the potential substitutional defect pairs in NpO_2_, this paper systematically examines the relationship between the formation energies of different valence states of substitutional impurity defects and the variation in Fermi energy under both oxygen-rich and anoxic conditions, as illustrated in Figure 8. The research findings indicate that under oxygen-rich conditions, it is difficult to form the substitutional impurity pair (O_Np_: Np_O_). Under anoxic conditions, however, two substitutional impurity pairs can form: {O_Np_^5+^: Np_O_^5−^} and {O_Np_^6+^: Np_O_^6−^}, with formation energies of 2.11 eV and 1.36 eV, respectively. Since the formation energies of these two substitutional impurity pairs are close to zero, it suggests their potential to exist within NpO_2_. The substitutional impurity defects can only form under anoxic conditions, and the relatively small difference in their formation energies indicates that the oxygen environment has a significant impact on the formation of substitutional impurity pairs, whereas the influence of valence state is less pronounced. The reason may be that O_Np_^6+^ requires adding O atoms to Np sites, but high *μ*O_2_ favors interstitial O over substitutional defects in oxygen-rich environment [36].

## 4. Conclusions

Utilizing first-principles plane-wave pseudopotential methods, this paper systematically investigates the intrinsic point defects that may exist in NpO_2_, as well as the effects of valence state, Fermi energy, and oxygen partial pressure on the formation energies of point defects. Furthermore, based on the formation of point defects, a systematic study of the existence and formation energies of Schottky, Frenkel, and substitutional impurity pairs in NpO_2_ is conducted. The research findings indicate that the most stable valence states of point defects vary with changes in the oxygen environment and Fermi energy. It is found that under oxygen-rich and anoxic conditions, the most stable point defects at the top of the valence band are V_Np_^4−^ and Np_O_^6+^, respectively, while at the bottom of the conduction band, they are O_Np_^6−^ and V_Np_^4−^, suggesting that substitutional impurity defects are worth investigating. Additionally, the study reveals that the oxygen environment is a key factor influencing the formation of defect pairs. Schottky, Frenkel, and substitutional impurity defect pairs are all challenging to form under oxygen-rich conditions. Under anoxic conditions, apart from the common point defect pairs, the existence of a new Schottky defect {2V_Np_^3−^: 3V_O_^2+^} is identified; there is a minor presence of Np-Frenkel defects {V_Np_^3−^: Np_i_^3+^} and {V_Np_^4+^: Np_i_^4+^}, as well as a minor presence of substitutional impurity pairs {O_Np_^5+^: Np_O_^5−^} and {O_Np_^6+^: Np_O_^6−^}. Based on thermodynamic analysis, the relative stability of defects in NpO_2_ is ordered as Schottky defects > O-Frenkel defects > Np-Frenkel defects > substitutional impurity pairs, and the predominant defects in NpO_2_ are Schottky defects. The thermodynamic analysis confirms that Schottky defects dominate NpO_2_’s defect chemistry under anoxic conditions, with negative formation energies indicating spontaneous formation.

## Figures and Tables

**Figure 1 materials-18-02487-f001:**
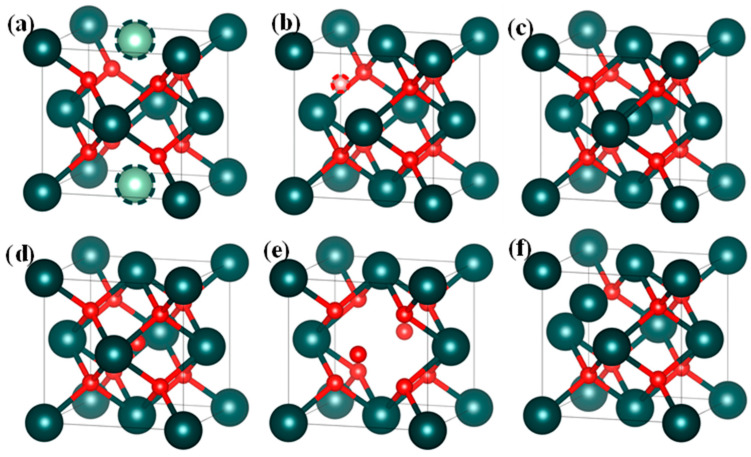
Schematic illustrations of the lattice structures for the six intrinsic point defects in NpO_2_: (**a**) O_Np_ (neptunium in the oxygen sublattice), (**b**) Np_O_ (oxygen in the neptunium sublattice), (**c**) Np_i_ (interstitial neptunium atom), (**d**) O_i_ (interstitial oxygen atom), (**e**) V_Np_ (vacancy on the neptunium site), (**f**) V_O_ (vacancy on the oxygen site).

**Figure 2 materials-18-02487-f002:**
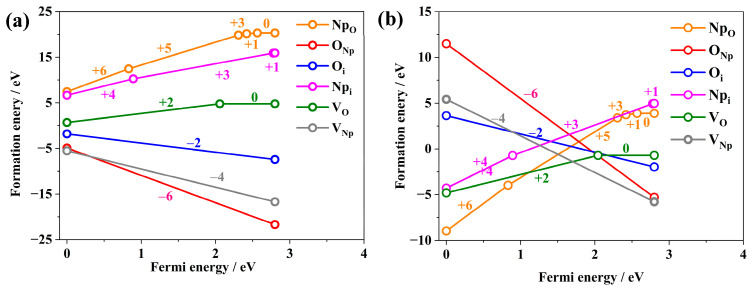
Variation in intrinsic point defect energies in NpO_2_ with respect to Fermi energy: (**a**) under oxygen-rich conditions, (**b**) under anoxic conditions.

**Figure 3 materials-18-02487-f003:**
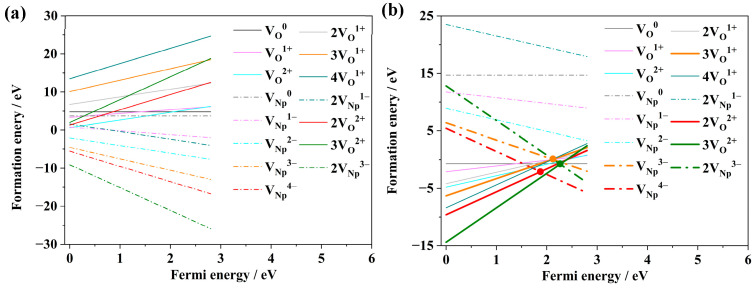
Variation in formation energies for different valence state vacancy defects in NpO_2_ with respect to Fermi energy: (**a**) under oxygen-rich conditions, (**b**) under anoxic conditions.

**Figure 4 materials-18-02487-f004:**
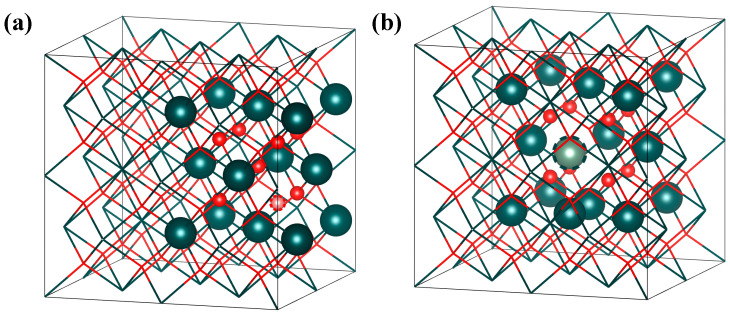
Frenkel defects in NpO_2_ when the interstitial atom and vacancy are at a 1NN (first nearest neighbor) distance: (**a**) O-Frenkel, (**b**) Np-Frenkel.

**Figure 5 materials-18-02487-f005:**
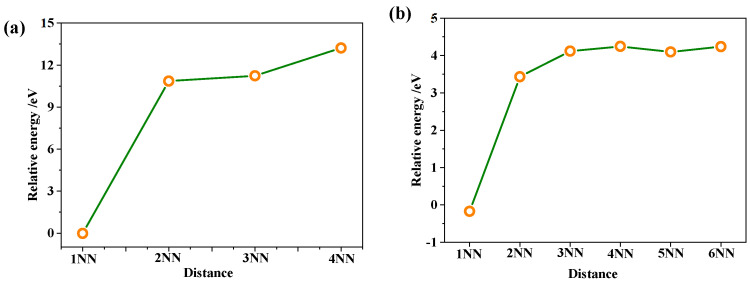
Relationship between the formation energy of Frenkel defects and the distance between vacancies and interstitial atoms in NpO_2_: (**a**) under anoxic conditions, (**b**) under oxygen-rich conditions.

**Figure 6 materials-18-02487-f006:**
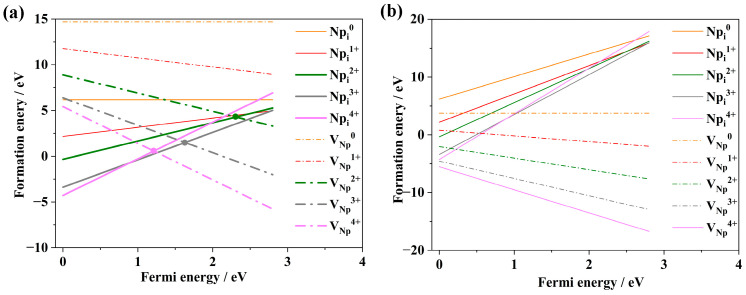
Relationship between point defects and Fermi energy in Np-Frenkel defects, along with formation energies: (**a**) under anoxic conditions, (**b**) under oxygen-rich conditions.

**Figure 7 materials-18-02487-f007:**
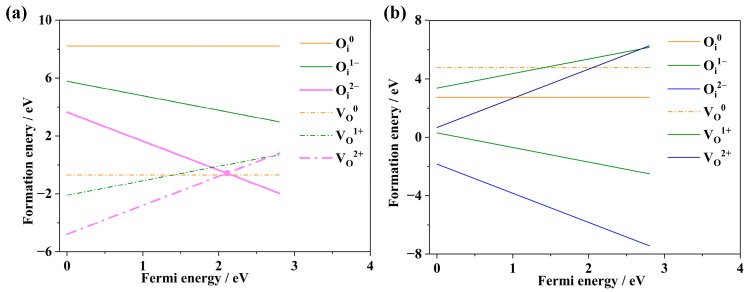
Relationship between point defects and Fermi energy in O-Frenkel defects, along with formation energies: (**a**) under anoxic conditions, (**b**) under oxygen-rich conditions.

**Figure 8 materials-18-02487-f008:**
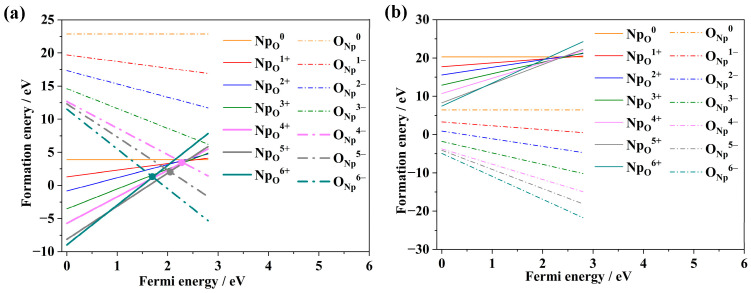
Relationship between point defects and Fermi energy in substitutional impurity defects, along with formation energies: (**a**) under anoxic conditions, (**b**) under oxygen-rich conditions.

## Data Availability

The original contributions presented in the study are included in the article. Further inquiries can be directed to the corresponding author.

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
