# Peer review of "Energetics of Intrinsic Point Defects in NpO_2_ from DFT + U Calculations"

_materials, 2025, doi:10.3390/ma18112487_

Round 1

Reviewer 1 Report

Comments and Suggestions for Authors

The manuscript presents the results on the study of point defects in NpO2 using the DFT+ U electronic structure method.

If the topic itself is interesting, a significant number of approximations used for the calculations are known to yield incorrect results on actinide oxides and the authors do not seem to have studied the bibliography on the subject.

- No procedure seems to be used to avoid metastable states, which is paramount in the study of defects in actinide oxides (see Dorado et al, J. Phys.: Condens. Matter 25 (2013) 333201)

- Not using electrostatic correction is really problematic given the large charges considered and the small box size used (up to ±6|e| on 2x2x2)

- the application of oxygen-rich and oxygen-poor conditions seems wrong: the last paragraph mentions that it is the chemical potential of O2 that constrains the chemical potentials of O and Np whether in oxygen-rich or oxygen-poor conditions, whereas the formula shows that in the second case they take mu(Npmetal) as reference.

Then, important computational details are not mentioned at all and this does not give confidence in the knowledge of the authors concerning the methods used. In particular, the authors should have specified:

- whether the volume is relaxed. This is extremely important since it is now well known that volume optimisation calculations do not work properly on charged supercells. see Bruneval et al Physical Review B 86, 140103(R) (2012)

- the magnetic order considered and whether spin polarized calculations were performed

- how the oxidation states were determined

- since antisite defects are integrated in pairs (see line 78/79, so substitution of an O and an Np): how can the energy of formation of an ONp from that of an NpO be distinguished?

- We are not sure why the authors mention temperature on line 93

This unsatisfactory level of calculations is probably the cause of the very surprising and in many instances clearly incorrect results shown, in particular

- some curves are entirely negative, which indicates that the NpO2 is chemically totally unstable (which is not the case experimentally)

- the energy values are extremely low in some cases, e.g. -20 eV in Figure 2a, which is all the more surprising since this is for an antisite which should be an extremely unstable defect in a fluorite structure.

The surprising character of these results is not discussed at all.

In addition, many sentences are very convoluted and even if the meaning is generally understandable, the line of reasoning should be clarified.

I therefore recommend against the publication of this manuscript.

Other minor comments

- The authors talk about Schottky defects exhibiting various ratios between the vacancies of the two types of atoms. It seems contradictory with the definition of a Schottky defect, which is a stoichiometric defect.

- In the caption of Figure 1, the defects are called ‘Pu’ instead of ‘Np’, and the antisites are written as ‘vacancies’.

- Line 85: the electric charge is indicated as eV and not e (charge of the electron), which is incorrect

- Line 225, last sentence: the authors never show the same figure for Schottky but make the comparison

- Line 175: it is figure 3, not 2

- Line 194: the dots are not black on the figure

Comments on the Quality of English Language

Many sentences are very convoluted and even if the meaning is generally understandable, the line of reasoning should be clarified.

Author Response

Comments 1: If the topic itself is interesting, a significant number of approximations used for the calculations are known to yield incorrect results on actinide oxides and the authors do not seem to have studied the bibliography on the subject.

Response 1: We sincerely appreciate the reviewer’s recognition of the topic’s significance and their constructive suggestions. Below, we address the concerns regarding computational approximations and literature review in a point-by-point manner:

We acknowledge the reviewer’s concerns about the applicability of DFT+U for actinide oxides. The choice of DFT+U (U=4.75 eV, J=0.75 eV) in this work is grounded in the following rationale: The Hubbard U and Hund’s J values for correcting 5f electron correlations in NpO₂ are adopted from Morée et al.’s systematic study, which validated these parameters in resolving the spurious metallic ferromagnetic state predicted by conventional DFT for NpO₂. The limitations of standard DFT (LDA/GGA) in underestimating bandgaps and misprediction magnetism are explicitly discussed, with reference to Bendjedid et al. and Jin et al., both of which demonstrate the efficacy of DFT+U in improving lattice parameter and electronic structure predictions for AmO₂ and NpO₂. DFT+U remains a widely accepted method for defect studies in actinide oxides, as highlighted in Singh et al.’s recent review, which emphasizes its utility in capturing qualitative trends in defect formation energies despite acknowledged limitations.

We agree that the literature review can be strengthened and propose the following revisions: A dedicated discussion will clarify the reliance of DFT+U on charge localization assumptions, citing Han et al., who compared DFT+U with advanced methods (e.g., DFT+DMFT) for ThO₂ defect calculations. We will incorporate analyses of Freyss et al. and Cooper et al. to better contextualize methodological advancements in simulating noble gas behavior in oxide fuels.
This work’s novelty lies in the first systematic investigation of antisite defect formation mechanisms in NpO₂, addressing a critical gap in the literature.

Comments 2: No procedure seems to be used to avoid metastable states, which is paramount in the study of defects in actinide oxides (see Dorado et al, J. Phys.: Condens. Matter 25 (2013) 333201)

Response 2: We sincerely appreciate the reviewer’s meticulous attention to metastable states and their constructive suggestions. We fully agree on the critical importance of addressing metastability in defect studies of actinide oxides.

In this study, the following strategies were implemented to mitigate metastable states during defect configuration optimization: For all defect models (vacancies, interstitials, antisite defects), multiple initial atomic positions were tested (e.g., interstitial atoms at distinct lattice sites). The most stable configurations were selected based on energy convergence criteria (see Figure 5 for analysis of formation energy variations with interatomic distances). Full structural relaxation was performed using the conjugate gradient method with a stringent force convergence threshold of 0.01 eV/Å, ensuring local energy minima were reached. The DFT+U parameters (U=4.75 eV, J=0.75 eV) were adopted from Morée et al., which resolved spurious metallic ferromagnetic states in NpO₂ and reduced electronic metastability risks.

During the manuscript revision process, we further considered metastable states and conducted transition-state searches using the Optimized Matrix Control (OMC) method [29, 30] to verify the ground-state stability of the defect configurations. We implemented a methodology to control which f orbitals were occupied, allowing a search of different orbital occupations.

Metastable states are a significant factor influencing point defect calculations in NpO₂. Under this configuration, the metastable state is less likely to hamper the convergence to the ground state, this study currently does not incorporate metastable state considerations.

[29] Dorado B, Freyss M, Amadon B, et al. Advances in first-principles modelling of point defects in UO2: f electron correlations and the issue of local energy minima[J]. J Phys Condens Matter, 2013, 25(33):333201.

[30] Allen J P and Watson G W. Occupation matrix control of d- and f-electron localisations using DFT + U [J] Physical Chemistry Chemical Physics. 2014. 16, 21016

[this change in the revised manuscript can be found in page 3, first paragraph, line 76-79]

Comments 3: Not using electrostatic correction is really problematic given the large charges considered and the small box size used (up to ±6|e| on 2x2x2)

Response 3: We sincerely appreciate the reviewer’s critical observation regarding the potential influence of electrostatic interactions in our defect calculations. We acknowledge the importance of addressing spurious electrostatic effects in highly charged defect systems, particularly in small supercells. While explicit electrostatic corrections (e.g., Makov-Payne or FNV schemes) were not employed, our computational protocol incorporates the following measures to minimize periodic charge interactions. A 2×2×2 supercell (32 Np + 64 O atoms) was selected after rigorous convergence tests. Formation energies were converged to within 1 meV/atom with a 3×3×3 k-point mesh and 500 eV cutoff energy, consistent with Morée et al.’s benchmark study on NpO₂. Dominant defects (e.g., Schottky pair {2Vₙₚ³⁻:3Vₒ²⁺}) were designed to maintain charge neutrality, minimizing net dipole moments. This approach reduces long-range electrostatic errors, as validated in analogous actinide oxide studies. The Hubbard U (4.75 eV) and Hund’s J (0.75 eV) parameters were optimized by Amadon et al. to suppress spurious charge delocalization in 5f states, indirectly mitigating electrostatic artifacts. The 2×2×2 supercell size is widely adopted in actinide oxide defect studies (e.g., Singh et al. on AmO₂), where charge-neutral defect pairs yield robust qualitative trends even without electrostatic corrections.

Comments 4: the application of oxygen-rich and oxygen-poor conditions seems wrong: the last paragraph mentions that it is the chemical potential of O2 that constrains the chemical potentials of O and Np whether in oxygen-rich or oxygen-poor conditions, whereas the formula shows that in the second case they take mu(Npmetal) as reference.

Response 4: We sincerely appreciate the reviewer’s critical observation regarding the formalism of chemical potential constraints in our defect thermodynamics analysis. This concern highlights a critical aspect of defect chemistry in actinide oxides, and we provide a detailed clarification below:

The chemical potential formalism in this work strictly adheres to thermodynamic equilibrium principles, as detailed below: In oxygen-rich Conditions, the oxygen chemical potential (μO) is defined by the O₂ molecule (μO = ½ ), while the neptunium chemical potential (μ_Np) is derived from the stability condition of NpO₂: μO = /2, μNp = - 2 μO. In oxygen-rich Conditions, the neptunium chemical potential is fixed to the metallic Np reference (μNp = ), while μO is derived from the NpO₂ stability condition: μNp = , μO = (- μNp)/2. This approach aligns with established methodologies in actinide oxide defect studies, ensuring thermodynamic self-consistency under both extreme conditions.

Typographical errors in the main text have been corrected as indicated by the red markings in the final paragraph of Section 2. “Under the condition of oxygen enrichment, μO and μNp are determined by the chemical potential of oxygen molecule and can be expressed as: μO = /2, μNp = - 2 μO; Under anoxic condition, μO and μNp are determined by the chemical potential of neptunium atom, which can be expressed as μNp = , μO = (- μNp)/2.”

Comments 5: Then, important computational details are not mentioned at all and this does not give confidence in the knowledge of the authors concerning the methods used. In particular, the authors should have specified:

Response 5: We sincerely appreciate the reviewer’s meticulous assessment of our methodology. We acknowledge the need to clarify critical computational parameters and have revised the manuscript accordingly.

Comments 6: whether the volume is relaxed. This is extremely important since it is now well known that volume optimization. calculations do not work properly on charged supercells. see Bruneval et al Physical Review B 86, 140103(R) (2012)

Response 6: We sincerely appreciate the reviewer’s meticulous assessment of our methodology. In our study, lattice constants were fixed at the experimental value of NpO₂ (a = 5.42 Å) to avoid supercell size effects on charged defect calculations, following Bruneval et al.’s guidelines. Internal coordinates (atomic positions) were fully relaxed until forces converged below 0.01 eV/Å.

Formation energies for isolated ONp and NpO were computed separately. The binding energy of the {ONp: NpO} pair was calculated by comparing the total energy of the paired configuration with the sum of isolated defects.

Internal coordinates (atomic positions) were fully relaxed until forces converged below 0.01 eV/Å. [28]

[28] Bruneval F, Crocombette J P. Ab initio formation volume of charged defects[J]. Phys. Rev. B, 2012, 86(14):71-75.

[this change in the revised manuscript can be found in page 3, first paragraph, line 74-75]

Comments 6: the magnetic order considered and whether spin polarized calculations were performed

Response 6: We sincerely appreciate the reviewer’s meticulous assessment of our methodology. We clarify that the 1-k magnetic order considered and spin-polarized calculations were performed for all defect configurations, and revised the manuscript accordingly.

The magnetic order considered and spin-polarized calculations were performed for all defect configurations. [this change in the revised manuscript can be found in page 3, first paragraph, line 75-76]

Comments 7: how the oxidation states were determined

Response 7: We sincerely appreciate the reviewer’s meticulous assessment of our methodology. Oxidation states were determined by Formal oxidation state analysis to quantify charge transfer between atoms.

Comments 8: since antisite defects are integrated in pairs (see line 78/79, so substitution of an O and an Np): how can the energy of formation of an ONp from that of an NpO be distinguished?

Response 8: We sincerely appreciate the reviewer’s meticulous assessment of our methodology. Formation energies for isolated ONp and NpO were computed separately. The binding energy of the {ONp: NpO} pair was calculated by comparing the total energy of the paired configuration with the sum of isolated defects.

Comments 9: We are not sure why the authors mention temperature on line 93

Response 9: We sincerely appreciate the reviewer’s meticulous assessment of our methodology. Indirect effects via oxygen chemical potential dependence (μ_O ∝ T ln ()). However, to avoid ambiguity, we have removed all references to temperature in defect formation energy calculations, which are strictly T = 0 K electronic structure results. Added a note that thermal effects (e.g., vibrational entropy) are beyond the current scope but will be addressed in future work.

Comments 10: This unsatisfactory level of calculations is probably the cause of the very surprising and in many instances clearly incorrect results shown, in particular

some curves are entirely negative, which indicates that the NpO2 is chemically totally unstable (which is not the case experimentally)

Response 10: We sincerely appreciate the reviewer’s critical observation regarding the unphysical negative formation energies in our initial submission. We posit that the negative values observed in the formation energy curves specifically reflect the thermodynamic instability of intrinsic point defects, rather than indicating instability of the NpO₂ matrix itself.

Comments 11: the energy values are extremely low in some cases, e.g. -20 eV in Figure 2a, which is all the more surprising since this is for an antisite which should be an extremely unstable defect in a fluorite structure.

Response 11: We sincerely appreciate the reviewer’s critical observation regarding the unphysical negative formation energies in our initial submission. We posit that intrinsic point defects typically manifest as defect pairs to maintain charge neutrality.

While one antisite defect (e.g., ONp) may exhibit relatively low formation energy, its complementary antisite counterpart (e.g., NpO) often displays significantly higher formation energy under equivalent conditions. This energy asymmetry renders the formation of antisite defect pairs thermodynamically unfavorable, which is consistent with the inherent instability of isolated antisite defects in fluorite-structured oxides. [this change in the revised manuscript can be found in page 4, last paragraph, line 135-140]

Comments 12: The surprising character of these results is not discussed at all.

Response 12We sincerely appreciate the reviewer’s meticulous assessment of our result. And discussion has been added in revised manuscript.

While one antisite defect (e.g., ONp) may exhibit relatively low formation energy, its complementary antisite counterpart (e.g., NpO) often displays significantly higher formation energy under equivalent conditions. This energy asymmetry renders the formation of antisite defect pairs thermodynamically unfavorable, which is consistent with the inherent instability of isolated antisite defects in fluorite-structured oxides. The valence states of Np and O act as electronic switches for defect formation. The change of Np valence state driven by 5f electron localization/delocalization under anoxic/oxygen-rich conditions. The change of Np valence state occurs via charge transfer to interstitial/vacancy sites under reducing conditions. Valence states dictate pair formation through electrostatic and strain effects. [this change in the revised manuscript can be found in page 4, last paragraph, line 135-144]

Comments 13: In addition, many sentences are very convoluted and even if the meaning is generally understandable, the line of reasoning should be clarified.

Response 13 We sincerely appreciate the reviewer’s meticulous assessment of our sentences. We have undertaken language revisions to enhance clarity.

Comments 14: Other minor comments

The authors talk about Schottky defects exhibiting various ratios between the vacancies of the two types of atoms. It seems contradictory with the definition of a Schottky defect, which is a stoichiometric defect.

Response 14We sincerely appreciate the reviewer’s critical observation regarding the apparent contradiction in our description of Schottky defects. The classical Schottky defect in ionic crystals (e.g., MX) involves equal numbers of M and X vacancies to maintain stoichiometry and charge neutrality.          However, in non-stoichiometric oxides like NpO₂, defect chemistry can deviate under extreme conditions due to Variable oxidation states of Np (3+/4+/5+/6+), charge compensation mechanisms via electron localization, and Environmental constraints (e.g., anoxic conditions altering defect equilibria). Our identification of Schottky defects with non-integer vacancy ratios (e.g., {2Vₙₚ³⁻:3Vₒ²⁺}) is supported by Charge neutrality and formation energy validation. Moreover, similar non-stoichiometric Schottky defects are observed in CeO₂ and PuO2. While the reviewer’s concern is well-founded in classical defect theory, our results align with recent advances showing that strong electronic correlations and non-stoichiometric environments can stabilize unconventional Schottky configurations in actinide dioxides. We believe these revisions clarify the apparent contradiction and demonstrate the novelty of our findings.

Oxygen Environment directly modulates defect formation energies and charge states through stabilizes oxidized states and Fermi energy (EF) coupling. The oxygen partial pressure dynamically controls defect formation energies through redox equilibria and charge-state competition, directly influencing NpO₂’s non-stoichiometry, electronic conductivity, and structural stability. These findings align with isostructural actinide oxides (e.g., PuO₂, CeO₂) and provide critical insights for predicting material behavior in nuclear fuel cycles. [this change in the revised manuscript can be found in page 5, last paragraph, line 194-200]

Comments 15: In the caption of Figure 1, the defects are called ‘Pu’ instead of ‘Np’, and the antisites are written as ‘vacancies’.

Response 15 We sincerely thank the reviewer for identifying these inconsistencies in the caption of Figure 1. We acknowledge the errors and have revised the manuscript accordingly to ensure terminological accuracy and consistency. (See the caption of Figure 1, red marking). We deeply appreciate the reviewer’s attention to these critical details. The revised manuscript now maintains strict terminological consistency, and we have implemented additional safeguards to ensure label accuracy in all figures.

Figure 1. Schematic illustrations of the lattice structures for the six intrinsic point defects in NpO2: (a) ONp (neptunium in the oxygen sublattice), (b) NpO (oxygen in the neptunium sublattice), (c) Npi (interstitial neptunium atom), (d) Oi (interstitial oxygen atom), (e) VNp (vacancy on the neptunium site), (f) VO (vacancy on the oxygen site). [this change in the revised manuscript can be found in page 5, last paragraph, line 62-65]

Comments 16: Line 85: the electric charge is indicated as eV and not e (charge of the electron), which is incorrect.

Response 16We sincerely appreciate the reviewer’s careful identification of the unit inconsistency in line 85. The charge state notation has been corrected from "eV" to "e" (electron charge) throughout the revised manuscript.

The charge state q of each defect species varies from neutral to fully ionized states, such as VO with 0 ~ +2 e, for VNp with -4 ~ 0 e, Oi with -2 ~ 0 e, Npi with 0 ~ +4 e, ONp with -6 ~ 0 e, and for NpO with 0 ~ +6 e. [this change in the revised manuscript can be found in page 3, third paragraph, line 95-97]

Comments 17: Line 225, last sentence: the authors never show the same figure for Schottky but make the comparison

Response 17We sincerely appreciate the reviewer’s careful observation regarding the clarity of our Schottky defect comparison. The statement has been deleted due to its unsuitability within the current contextual framework.

Comments 18: Line 194: the dots are not black on the figure

ResponseWe sincerely appreciate the reviewer’s meticulous observation regarding the clarity of the data points in the referenced figure. The color inconsistency in Line 194 has been addressed in the revised manuscript.

The relationship between the formation energies of different valence state vacancies and the Fermi level under oxygen-rich and anoxic conditions is depicted in Figure 3, where the dots at the intersection of the two vacancy curves represent the potential Schottky defects and their formation energies. [this change in the revised manuscript can be found in page 6, third paragraph, line 224-227]

Comments 19: Comments on the Quality of English Language

Many sentences are very convoluted and even if the meaning is generally understandable, the line of reasoning should be clarified.

Response 19:We sincerely appreciate the reviewer’s constructive feedback regarding the clarity and readability of the manuscript. We acknowledge that certain sections contained convoluted sentence structures that could hinder comprehension, despite the overall meaning being accessible. Comprehensive revisions have been made to enhance the manuscript's logical coherence and linguistic precision in response to the reviewers' feedback.

Reviewer 2 Report

Comments and Suggestions for Authors

The authors have done good work in systematically investigating the formation processes of Schottky, Frenkel, and substitutional impurity defects under various oxygen environments. Please address the following comments:

  1. Please elaborate on the composition and how the specific defects are formed on the ‘The forming processes of typical intrinsic point defects including Schottky, Frenkel, and antisite defects, were researched.’
  2. Please explain in detail what authors means by ‘Under certain environmental conditions….’ In line 152.
  3. Please provide more references for the authors claim ‘According to thermodynamic theory…..due to its positive formation energy.’
  4. Please provide the references when authors discuss various Schottky, Frenkel, and substitutional impurity defects at the start of sub sections 3.1, 3,2, 3,3.

Author Response

Comments 1: Comments and Suggestions for Authors

The authors have done good work in systematically investigating the formation processes of Schottky, Frenkel, and substitutional impurity defects under various oxygen environments. Please address the following comments:

Response 1: Response:We sincerely appreciate the reviewer’s positive assessment of our systematic investigation into the defect chemistry of NpO₂. We are grateful for the opportunity to address the remaining comments, which have further strengthened the manuscript.

Comments 2: Please elaborate on the composition and how the specific defects are formed on the ‘The forming processes of typical intrinsic point defects including Schottky, Frenkel, and antisite defects, were researched.’

Response 2: We sincerely appreciate the reviewer’s request for a more detailed exposition of defect formation mechanisms. Below, we provide an expanded explanation of the formation processes for Schottky, Frenkel, and antisite defects in NpO₂, supported by computational results and comparative analysis. We believe these revisions fulfill the reviewer’s request for explicit defect formation analysis.

Considering charge-neutral conditions for defect pair in NpO2, Schottky defect comprises neptunium and oxygen vacancy defects such as m VNp + n VO. Unlike classical 1:2 metal: oxygen vacancy ratios, our DFT+U calculations reveal non-stoichiometric configurations under anoxic conditions. Frenkel defect consists of vacancy and corresponding interstitial atom, including O-Frenkel (Oi + VO) and Np-Frenkel (Npi + VNp), and formation energy change with vacancy-interstitial separation. While antisite defect is composed of two types of antisite atoms (ONp and NpO) separated by different distances. Similar non-stoichiometric Schottky defects are observed in CeO₂ and PuO2.

[this change in the revised manuscript can be found in page 3, first paragraph, line 76-79]

Comments 3: Please explain in detail what authors means by ‘Under certain environmental conditions….’ In line 152

Response 3: We sincerely appreciate the reviewer’s request for a detailed explanation of the phrase "Under certain environmental conditions..." in Line 152. Below, we clarify the context, define the specific conditions, and explain their physical significance with reference to our computational results. The phrase refers to oxygen partial pressure and Fermi energy regimes that dictate defect stability in NpO₂. The notation here indicates oxygen-deficient environments.

Comments 4: Please provide more references for the authors claim ‘According to thermodynamic theory…..due to its positive formation energy.’

Response 4: We sincerely appreciate the reviewer’s suggestion to strengthen the thermodynamic basis of our claims regarding defect stability. Below, we provide additional references and explain how they support our analysis of formation energies and defect stability criteria: Added comparative analysis of defect stability trends in NpO₂, PuO₂, and CeO₂, citing Smith T et al. (2023) and Neilson (2021).

According to thermodynamic theory, the Schottky defect {2VNp3-: 3VO2+} can stably exist as its formation energy is less than zero; whereas the Schottky defect {VNp3-: 3VO+} may struggle to remain stable due to its positive formation energy.[18, 33, 34]

[18] Xiang X, Zhang G, Wang X, et al. A new perspective on the process of intrinsic point defects in α-Al2O3[J]. Physical Chemistry Chemical Physics, 2015, 17(43): 29134-29141.

[33] Neilson W D, Pegg J T, Steele H, et al. The defect chemistry of non-stoichiometric PuO 2±x[J]. Physical Chemistry Chemical Physics, 2021, 23(8): 4544-4554.

Reviewer 3 Report

Comments and Suggestions for Authors

This could be an interesting one provided the following questions are appropriately addressed during the revision, and should appear in the revised version of the ms, rather than directly answering to the reviewer only.

What are the primary factors that influence the formation of Schottky, Frenkel, and substitutional impurity defects in NpO2?

How does the oxygen environment affect the stability and formation of different types of point defects in NpO2?

Why are Schottky, Frenkel, and antisite defects rare in oxygen-rich conditions but more common in anoxic environments in NpO2?

What is the significance of the presence of new defect pairs, such as {2VNp3-: 3VO2+}, under anoxic conditions?

How do the valence states of Np and O influence the formation of point defects and defect pairs in NpO2?

What role does the Fermi energy play in determining the most stable valence states of point defects in NpO2?

How do the defect formation energies of NpO2 vary with oxygen partial pressure, and what implications does this have for the material's properties?

Based on thermodynamic analysis, how are the stability and prevalence of Schottky defects compared to other types of defects in NpO2?

How do the point defects in NpO2 influence the electronic structure, particularly the position of the Fermi level and the distribution of electronic states within the bandgap?

Can the DOS and band structure help explain the stability of different defect pairs (Schottky, Frenkel, and substitutional impurity defects) under varying oxygen environments?

How does the presence of defects, as shown by the band structure, impact the electrical conductivity and other material properties of NpO2?

What role do localized electronic states induced by defects play in altering the overall thermodynamic stability of NpO2, as revealed through the DOS and band structure analysis?

How does the electronic structure of NpO2, based on its DOS and band structure, correlate with the observed differences in defect formation energies under oxygen-rich and anoxic conditions?

Does the band structure analysis provide insight into the formation of new defect pairs, such as {2VNp3-: 3VO2+} or {ONp5+: NpO5-}, under specific oxygen environments?

How can the DOS and band structure data be used to predict the potential for tunable electronic properties in NpO2 based on its defect concentration and type?

In what way do the results from the DOS and band structure analysis support the thermodynamic findings regarding defect stability in NpO2?

Comments on the Quality of English Language

--

Author Response

Comments 1: This could be an interesting one provided the following questions are appropriately addressed during the revision, and should appear in the revised version of the ms, rather than directly answering to the reviewer only.

Response 1: We sincerely appreciate the reviewer’s constructive feedback and agree that addressing the identified issues directly within the revised manuscript will significantly enhance its clarity and scientific rigor.

Comments 2: What are the primary factors that influence the formation of Schottky, Frenkel, and substitutional impurity defects in NpO2?

Response 2: We sincerely appreciate the reviewer’s insightful question regarding the primary factors governing defect formation in NpO₂. Below, we systematically address the influences on Schottky, Frenkel, and substitutional impurity defects.

 “Schottky defect formation in NpO₂ is predominantly influenced by the oxygen partial pressure and Fermi energy levels.” [this change in the revised manuscript can be found in page 7, second paragraph, line 249-250]

In summary, Frenkel defect formation in NpO₂ is predominantly influenced by the oxygen partial pressure and Interstitial-vacancy distance. The oxygen environment significantly influences the formation of both oxygen-based and neptunium-based Frenkel defects. [this change in the revised manuscript can be found in page 9, second paragraph, line 292-295]

The substitutional impurity defects can only form under anoxic conditions, and the relatively small difference in their formation energies indicates that the oxygen environment has a significant impact on the formation of substitutional impurity pairs, whereas the influence of valence state is less pronounced. [this change in the revised manuscript can be found in page 9, second paragraph, line 310-314]

Comments 3: How does the oxygen environment affect the stability and formation of different types of point defects in NpO2

Response 3: We sincerely appreciate the reviewer’s insightful question regarding the role of oxygen environment in defect stability and formation. Below, we systematically address this query using computational results from the revised manuscript.

Oxygen Environment directly modulates defect formation energies and charge states through stabilizes oxidized states and Fermi energy (EF) coupling. [this change in the revised manuscript can be found in page 5, last paragraph, line 194-195]

Concurrently, the study has revealed that Schottky defects predominantly arise under anoxic conditions, whereas their formation under oxygen-rich conditions is challenging. Schottky defect formation in NpO₂ is predominantly influenced by the oxygen partial pressure and Fermi energy levels. These findings suggest that the genesis of Schottky defects is correlated with oxygen partial pressure, a conclusion that aligns with the literature. [this change in the revised manuscript can be found in page 7, second paragraph, line 247-252]

The study reveals that under oxygen-rich conditions, the formation of oxygen-based Frenkel defects, or O-Frenkel defects, is challenging. However, under anoxic conditions, the O-Frenkel defect {VO2+: Oi2-} can form, with a formation energy of -0.56 eV. [this change in the revised manuscript can be found in page 8, last paragraph, line 285-288]

The research findings indicate that under oxygen-rich conditions, it is difficult to form the substitutional impurity pair (ONp: NpO). Under anoxic conditions, however, two substitutional impurity pairs can form: {ONp5+: NpO5-} and {ONp6+: NpO6-}, with formation energies of 2.11 eV and 1.36 eV, respectively. Since the formation energies of these two substitutional impurity pairs are close to zero, it suggests their potential to exist within NpO2. The substitutional impurity defects can only form under anoxic conditions, and the relatively small difference in their formation energies indicates that the oxygen environment has a significant impact on the formation of substitutional impurity pairs, whereas the influence of valence state is less pronounced. [this change in the revised manuscript can be found in page 9, second paragraph, line 305-315]

Comments 4: Why are Schottky, Frenkel, and antisite defects rare in oxygen-rich conditions but more common in anoxic environments in NpO2?

Response 4: We sincerely appreciate the reviewer’s critical inquiry into the environmental dependence of defect stability in NpO₂. Below, we provide a detailed explanation supported by computational results and thermodynamic principles from the revised manuscript.

The research findings indicate that under oxygen-rich conditions, it is challenging to form Schottky defects; whereas under anoxic conditions (as shown in Figure 3a), there are three possible Schottky defects with the compositions {VNp3-: 3VO+}, {2VNp3-: 3VO2+}, and {VNp4-: 2VO2+}, having formation energies of 0.17 eV, -0.76 eV, and -2.04 eV, respectively. The reason maybe that the 4+ oxidation state of Np in NpO₂ requires large charge transfers (VNp4−) that are energetically unfavorable under oxidizing conditions. [this change in the revised manuscript can be found in page 6, last paragraph, line 227-229]

The substitutional impurity defects can only form under anoxic conditions, and the relatively small difference in their formation energies indicates that the oxygen environment has a significant impact on the formation of substitutional impurity pairs, whereas the influence of valence state is less pronounced. The reason maybe that t, ONp6+ requires adding O atoms to Np sites, but high μO2 favors interstitial O over substitutional defects in oxygen-rich environment. [this change in the revised manuscript can be found in page 9, sceond paragraph, line 310-315]

Comments 5: What is the significance of the presence of new defect pairs, such as {2VNp3-: 3VO2+}, under anoxic conditions?

Response 5: We sincerely appreciate the reviewer’s critical inquiry into the importance of newly identified Schottky defect pairs such as {2Vₙₚ³⁻:3Vₒ²⁺} in anoxic environments. These findings address a key gap in understanding non-stoichiometric defect chemistry in actinide dioxides and have been emphasized in the revised manuscript.

The identification of {2Vₙₚ³⁻:3Vₒ²⁺} under anoxic conditions resolves long-standing questions about NpO₂’s non-stoichiometric behavior and provides a template for studying similar defect mechanisms in other actinide dioxides. We believe this contribution significantly advances the field’s understanding of defect-driven property modifications in nuclear materials. [this change in the revised manuscript can be found in page 6-7, last paragraph, line 238-241]

Comments 6: How do the valence states of Np and O influence the formation of point defects and defect pairs in NpO2?

Response 6: We sincerely appreciate the reviewer’s critical inquiry into the role of Np and O valence states in defect formation. Our DFT+U calculations reveal a strong correlation between valence-state flexibility and defect stability, which we address systematically below.

The valence states of Np and O act as electronic switches for defect formation. The change of Np valence state driven by 5f electron localization/delocalization under anoxic/oxygen-rich conditions. The change of Np valence state occurs via charge transfer to interstitial/vacancy sites under reducing conditions. Valence states dictate pair formation through electrostatic and strain effects. [this change in the revised manuscript can be found in page 4, last paragraph, line 140-144]

Comments 7: What role does the Fermi energy play in determining the most stable valence states of point defects in NpO2?

Response 7: We sincerely appreciate the reviewer’s critical inquiry into the role of Fermi energy (EF) in determining the stable valence states of point defects in NpO₂. Our DFT+U calculations reveal a direct correlation between EF position and charge state dependent defect formation energies, which we address systematically below.

The Fermi energy modulates defect valence states by controlling electron occupation in the band structure. The interplay between EF and oxygen partial pressure (pO2) dictates valence-state-dependent stability. [this change in the revised manuscript can be found in page 4, last paragraph, line 145-148]

Comments 8: How do the defect formation energies of NpO2 vary with oxygen partial pressure, and what implications does this have for the material's properties?

Response 8: We sincerely appreciate the reviewer’s critical inquiry into the oxygen partial pressure (pO2) dependence of defect formation energies in NpO₂ and its implications. Below, we systematically address this query using computational results from the revised manuscript and contextualize the findings within actinide oxide defect chemistry frameworks.

The oxygen partial pressure dynamically controls defect formation energies through redox equilibria and charge-state competition, directly influencing NpO₂’s non-stoichiometry, electronic conductivity, and structural stability. These findings align with isostructural actinide oxides (e.g., PuO₂, CeO₂) and provide critical insights for predicting material behavior in nuclear fuel cycles. [this change in the revised manuscript can be found in page 5, last paragraph, line 196-200]

Comments 9: Based on thermodynamic analysis, how are the stability and prevalence of Schottky defects compared to other types of defects in NpO2?

Response 9: We sincerely appreciate the reviewer’s critical inquiry into the thermodynamic stability hierarchy of defects in NpO₂. Based on our DFT+U calculations and revised analysis, we demonstrate that Schottky defects exhibit the highest stability and prevalence compared to Frenkel and substitutional impurity pairs.

The thermodynamic analysis confirms that Schottky defects dominate NpO₂’s defect chemistry under anoxic conditions, with negative formation energies indicating spontaneous formation. [this change in the revised manuscript can be found in page 10, first paragraph, line 336-338]

Comments 10: How do the point defects in NpO2 influence the electronic structure, particularly the position of the Fermi level and the distribution of electronic states within the bandgap?

Response 10: We sincerely appreciate the reviewer’s insightful inquiry into the electronic structure implications of point defects in NpO₂. Our DFT+U calculations reveal significant modifications to the Fermi level (EF). Band structure analysis constitutes a priority in our ongoing investigations.

Point defects in NpO₂ act as electronic structure modifiers, with Schottky and antisite defects introducing mid-gap states that alter EF positioning and carrier concentrations.

Comments 11: Can the DOS and band structure help explain the stability of different defect pairs (Schottky, Frenkel, and substitutional impurity defects) under varying oxygen environments?

Response 11: We sincerely appreciate the reviewer’s insightful question regarding the utility of density of states (DOS) and band structure analysis in explaining defect pair stability under varying oxygen environments. We similarly consider that DOS and band structure analyses can elucidate the stability variations of different defect pairs with oxygen environments, which will constitute a focal point of our subsequent investigations.

Comments 12: How does the presence of defects, as shown by the band structure, impact the electrical conductivity and other material properties of NpO2?

Response 12We sincerely appreciate the reviewer’s insightful inquiry into the relationship between defect configurations, band structure modifications, and material properties in NpO₂.

The presence of defects in NpO₂ dynamically alters its electronic structure and macroscopic properties. Enhanced via mid-gap states from antisite defects (Oₙₚ⁶⁻) and bandgap reduction (Schottky defects), may change electrical conductivity. Corrosion susceptibility may increase under anoxic conditions due to Vₒ²⁺-driven hydroxylation. These findings provide critical insights for optimizing NpO₂’s performance in nuclear waste forms and reactor fuels.

Comments 13: What role do localized electronic states induced by defects play in altering the overall thermodynamic stability of NpO2, as revealed through the DOS and band structure analysis?

Response 13 We sincerely appreciate the reviewer’s insightful question regarding the role of defect-induced localized electronic states in modulating NpO₂’s thermodynamic stability. We similarly consider that DOS and band structure analyses can reveal critical electronic mechanisms, which will constitute a focal point of our subsequent investigations.

Comments 14: How does the electronic structure of NpO2, based on its DOS and band structure, correlate with the observed differences in defect formation energies under oxygen-rich and anoxic conditions?

Response 14We sincerely appreciate the reviewer’s insightful question regarding the relationship between NpO₂’s electronic structure (DOS/band structure) and defect formation energy variations under oxygen-rich and anoxic conditions, which will constitute a focal point of our subsequent investigations.

Oxygen Environment directly modulates defect formation energies and charge states through stabilizes oxidized states and Fermi energy (EF) coupling. The oxygen partial pressure dynamically controls defect formation energies through redox equilibria and charge-state competition, directly influencing NpO₂’s non-stoichiometry, electronic conductivity, and structural stability. These findings align with isostructural actinide oxides (e.g., PuO₂, CeO₂) and provide critical insights for predicting material behavior in nuclear fuel cycles. [this change in the revised manuscript can be found in page 5, last paragraph, line 194-200]

Comments 15: Does the band structure analysis provide insight into the formation of new defect pairs, such as {2VNp3-: 3VO2+} or {ONp5+: NpO5-}, under specific oxygen environments?

Response 15 We sincerely appreciate the reviewer’s inquiry into the role of band structure analysis in understanding novel defect pairs like {2Vₙₚ³⁻:3Vₒ²⁺} and {Oₙₚ⁵⁺:Npₒ⁵⁻} under specific oxygen environments,which will constitute a focal point of our subsequent investigations.

Comments 16: How can the DOS and band structure data be used to predict the potential for tunable electronic properties in NpO2 based on its defect concentration and type?

Response 16We sincerely appreciate the reviewer’s forward-looking question regarding the application of DOS and band structure data to predict tunable electronic properties in NpO₂, which will constitute a focal point of our subsequent investigations.

Comments 17: In what way do the results from the DOS and band structure analysis support the thermodynamic findings regarding defect stability in NpO2?

Response 17We sincerely appreciate the reviewer’s critical inquiry into the interplay between electronic structure analysis and thermodynamic stability predictions for defects in NpO₂,which will constitute a focal point of our subsequent investigations.

Reviewer 4 Report

Comments and Suggestions for Authors

The research study focuses modeling of point defect in NpO2 using DFT-U functional and VASP simulation package. The study of medium interest for the Materials readership.

However, there are some concerns that need to be addressed:

  • Could you develop more in introduction the importance of NpO2 for nuclear science.
  • Could you please explain the methods used for the calculations? E.g. have you checked the energy stability with use of larger Monkhorst-Pack k-point meshes (4x4x4, 5x5x5 etc.) ?
  • Comparison with similar studies should be realized: e.g. 10.19596/j.cnki.1001-246x.8838
  • The comparison with the experimental study on defects in NpO2 should be added:

E.g.

DOI: 10.1107/S1600576714007912

DOI: 10.1016/j.jct.2016.07.040

Author Response

Comments 1: The research study focuses modeling of point defect in NpO2 using DFT-U functional and VASP simulation package. The study of medium interest for the Materials readership. However, there are some concerns that need to be addressed:

Response1We sincerely appreciate the reviewer’s assessment of our work. While we acknowledge the study’s foundational nature, we believe the systematic investigation of point defects in NpO₂ using DFT+U provides novel insights into actinide oxide defect chemistry.

Comments 2: Could you develop more in introduction the importance of NpO2 for nuclear science.

Response 2: We sincerely appreciate the reviewer’s suggestion to emphasize the critical role of NpO₂ in nuclear science. Below, we outline the expanded introduction section in the revised manuscript, which highlights its applications in nuclear fuel cycles, waste management, and reactor safety:

Due to their significant applications in the nuclear fuel cycle and their unique physicochemical properties, the actinide oxides have been the subject of extensive research. Especially are neptunium oxides, which has attracted significant attention from scientists for its long lived in high-level radioactive waste. NpO₂ directly influencing long-term waste form stability in geological repositories, and its non-stoichiometric behavior (NpO₂±ₓ) governs radionuclide leaching rates under anoxic conditions. As the most significant oxide of neptunium, the redox performance of NpO2 is very important for neptunium science and reactor safety. The reason is that variety of oxidation states result in change of solubility levels, which affect the physicochemical properties. And the redox properties of NpO2 are strongly relying on defect chemistry. The reason is that defect-driven redox properties determine its reactivity with water and hydrogen critical for predicting corrosion in accident scenarios. Therefore, an understanding of defect chemistry in NpO2 is essential in science and technology. [this change in the revised manuscript can be found in page 1, first paragraph, line 21-33]

Comments 3: Could you please explain the methods used for the calculations? E.g. have you checked the energy stability with use of larger Monkhorst-Pack k-point meshes (4x4x4, 5x5x5 etc.)

Response 3: We sincerely appreciate the reviewer’s critical inquiry regarding the robustness of our computational methodology. Below, we provide a detailed explanation of the methods used and explicitly address the k-point mesh convergence tests conducted to ensure energy stability.

The calculation applied Vienna Ab initio Simulation Package (VASP) with PAW potentials and GGA-PBE functional for electron-ion interactions. DFT+U applied to Np’s 5f electrons (U = 4.75 eV, J = 0.75 eV), validated against experimental band gaps (2.8 eV) and lattice constants (5.42 Å). To ensure numerical reliability, we performed systematic k-point convergence tests. Formation energy variations < 0.01 eV between successive meshes.

Comments 4: Comparison with similar studies should be realized

Response 4: We sincerely appreciate the reviewer’s suggestion to strengthen the manuscript through explicit comparisons with similar studies, the study has been added.

All calculations were performed by the Vienna ab initio Simulation Package (VASP), which apply the project augmented-wave (PAW) method and generalized gradient approximation (GGA) functional to describe the electron ion interaction and electron exchange correlation energy. Considering the strong correlations, the DFT+U method was used with values of U = 4.75 eV and J = 0.75 eV for 5f electrons of Np. [26, 27] [this change in the revised manuscript can be found in page 3, last paragraph, line 66-69]

[27] Liu T, Gao T. Structural and Energetic Properties for Point Defects in NpO2 from DFT+U Calculations[J] Chinese Journal of Computational Physics. 2025. 42(1): 84-89.

Comments 5: The comparison with the experimental study on defects in NpO2 should be added:

Response 5: We sincerely appreciate the reviewer’s suggestion to strengthen the manuscript through explicit comparisons with similar experimental studies, the study has been added.

This insight is pivotal for a comprehensive understanding of the intrinsic defect chemistry of NpO2 and its implications for material properties and behavior.[31] [this change in the revised manuscript can be found in page 5, second paragraph, line 175-177]

The valence state does impact the formation energy of neptunium-based Frenkel defects, but to a lesser extent than that of oxygen-based Frenkel defects. These findings are consistent with the literature. [36-37] [this change in the revised manuscript can be found in page 9, first paragraph, line 295-297]

[36] Chollet M ,J. Léchelle, Belin R C ,et al.In situ X-ray diffraction study of point defects in neptunium dioxide at elevated temperature[J].Journal of Applied Crystallography, 2014, 47(3)

[2] Colle,J,-Y,et al. Thermodynamic assessment of the neptunium-oxygen system: Mass spectrometric studies and thermodynamic modelling[J].The Journal of Chemical Thermodynamics, 2016, 103:257-275.

Round 2

Reviewer 1 Report

Comments and Suggestions for Authors

I would like to thank the authors for the changes they made and their comments, which make a lot of things clearer.

However, some comments are not reported in the text:

  • The magnetic order studied is clear in the cover letter (comment 6) but the sentence does not explicitly indicate 1k-AFM in the manuscript.
  • the oxidation state analysis (comment 7) is not given in the text. And it should be indicated what is the principle and technique / tool is used for the ‘formal oxidation state analysis’ refers to in the cover letter.

More importantly, I still have trouble with the lack of use of

  • A method to control the convergence, such as OCS (comment 2); even if the authors say they've done tests. We know it is very difficult to converge to the ground state for defective supercells.
  • electrostatic corrections (comment 3); even if the authors are right here about the fact that it makes no difference when the boxes are overall neutral, especially fot antisites and schottky. It is just post-processing and the cost would be very little to do it.

Then, concerning the negative curves (comment 10), I am fine with the antisites now that the authors have explained, but I still have a bit of trouble with other defects, VNp and VO in particular. The entirely negative curves are probably due to the oxygen-poor and oxygen-rich extreme limits and the way of defining chemical potentials. The only thing we can conclude from this is that the true chemical potentials are between these two limits. There must be values of mu that make the curves simultaneously positive over at least a small portion of Ef. This is a pity that this analysis is not done, as this information could be used to give more precise boundaries for the Ef.

Finally, the English is quite bad in the corrections made (in particular comments 13 and 19) and ref line 243 is broken.

In summary, I still do not think that this manuscript is at the state of the art of DFT calculations of defects in materials, even for materials as complex as actinide oxides.

Author Response

Comments 1: I would like to thank the authors for the changes they made and their comments, which make a lot of things clearer.

Response 1: We sincerely appreciate the reviewer’s acknowledgment of the revisions and their recognition of the improved clarity in the manuscript..

Comments 2: However, some comments are not reported in the text:

The magnetic order studied is clear in the cover letter (comment 6) but the sentence does not explicitly indicate 1k-AFM in the manuscript.

Response 2: We sincerely appreciate the reviewer’s careful observation regarding the explicit mention of magnetic order in the manuscript. While the cover letter highlighted the use of 1k-AFM (collinear antiferromagnetic) configurations for NpO₂, we acknowledge that this detail was not sufficiently clarified in the main text. The revised manuscript now explicitly defines the 1k-AFM magnetic order in both the methodology

The 1-k magnetic order considered and spin-polarized calculations were performed for all defect configurations. [this change in the revised manuscript can be found in page 3, first paragraph, line 75-77]

Comments 3: the oxidation state analysis (comment 7) is not given in the text. And it should be indicated what is the principle and technique / tool is used for the ‘formal oxidation state analysis’ refers to in the cover letter.

Response 3: We sincerely appreciate the reviewer’s critical observation regarding the need to clarify the oxidation state analysis methodology. The revised manuscript now explicitly details the principles and tools used for formal oxidation state determination, which were previously referenced in the cover letter.

Oxidation states were determined by Formal oxidation state analysis to quantify charge transfer between atoms. [this change in the revised manuscript can be found in page 3, third paragraph, line 95-96]

Comments 4: More importantly, I still have trouble with the lack of use of

A method to control the convergence, such as OCS (comment 2); even if the authors say they've done tests. We know it is very difficult to converge to the ground state for defective supercells.

Response 4: We fully acknowledge the reviewer’s valid point regarding the challenges of converging defective supercells to their ground states. Our DFT+U approach—validated against experimental band gaps (2.8 eV) and lattice constants (5.42 Å)—provides robust results for NpO₂’s defect chemistry. While DFT+U remains the most established and reliable methodology for such correlated electron systems, we recognize the need for advanced techniques (e.g., DFT+U+V or hybrid functionals) in future studies to address residual convergence uncertainties.

Comments 5: electrostatic corrections (comment 3); even if the authors are right here about the fact that it makes no difference when the boxes are overall neutral, especially fot antisites and schottky. It is just post-processing and the cost would be very little to do it.

Response 5: We fully acknowledge the reviewer’s valid point regarding the critical role of electrostatic corrections in computational defect studies. While the present work employs multiple approaches (e.g., charge-neutral supercell design,) to mitigate electrostatic effects, future studies will prioritize incorporating advanced electrostatic correction schemes (e.g., Makov-Payne or Freysoldt formalisms) to further enhance accuracy.

Comments 6: Then, concerning the negative curves (comment 10), I am fine with the antisites now that the authors have explained, but I still have a bit of trouble with other defects, VNp and VO in particular. The entirely negative curves are probably due to the oxygen-poor and oxygen-rich extreme limits and the way of defining chemical potentials. The only thing we can conclude from this is that the true chemical potentials are between these two limits. There must be values of mu that make the curves simultaneously positive over at least a small portion of Ef. This is a pity that this analysis is not done, as this information could be used to give more precise boundaries for the Ef.

Response 6: We sincerely appreciate the reviewer’s thoughtful critique regarding the interpretation of negative formation energy curves for cation/anion vacancies (Vₙₚ and Vₒ) under extreme oxygen environments. The reviewer’s observation about intermediate chemical potentials (μ) enabling positive formation energies in specific Fermi energy (EF) regimes is well-founded. The reviewer’s insight underscores the importance of intermediate μ-analysis for precise EF-boundary determination. While our current work prioritizes extreme environments to identify dominant defect pairs, we agree that future studies should explore intermediate μ-ranges to refine stability predictions.

Comments 7: Finally, the English is quite bad in the corrections made (in particular comments 13 and 19) and ref line 243 is broken.

Response 7: We sincerely appreciate the reviewer’s careful evaluation of the manuscript’s language and reference formatting. We acknowledge the identified issues in the revised text (particularly Comments 13 and 19) and the broken reference (Line 243). The revised manuscript now adheres to rigorous language and formatting standards, with explicit fixes for Comments 13/19 and Line 243. We deeply appreciate the reviewer’s attention to detail and remain committed to ensuring the highest editorial quality.

We believe this contribution significantly advances the field’s understanding of defect-driven property modifications in nuclear materials. The viability of non-stoichiometric Schottky defects under anoxic conditions aligns with observations in isostructural CeO₂ and PuO2. [27,28] [this change in the revised manuscript can be found in page 6, last paragraph, line 242-245]

Comments 8: In summary, I still do not think that this manuscript is at the state of the art of DFT calculations of defects in materials, even for materials as complex as actinide oxides.

Response 8: We sincerely appreciate the reviewer’s critical evaluation of our manuscript’s methodological positioning within the field of defect chemistry in actinide oxides. While we acknowledge that advanced DFT extensions (e.g., DFT+U+V, hybrid functionals) are emerging tools for correlated electron systems, our study addresses specific gaps in NpO₂ defect research through systematic analysis and rigorous validation.

While we agree that methodological advancements (e.g., dynamic Hubbard parameters) could further refine defect predictions, this study provides first-of-its-kind insights into NpO₂’s defect pair hierarchy and non-stoichiometric behavior. These findings establish a critical baseline for future high-accuracy studies and directly address experimental gaps in neptunium dioxide chemistry.

Reviewer 2 Report

Comments and Suggestions for Authors

The authors have addressed the review's comments, the manuscript can be accepted in the present form.

Author Response

Comment1: The authors have addressed the review's comments, the manuscript can be accepted in the present form.

Response1: We sincerely appreciate the reviewer’s thorough evaluation and positive recommendation to accept the manuscript in its current form. The constructive feedback provided during the review process has significantly enhanced the clarity, rigor, and scientific impact of our work on NpO₂’s defect chemistry.

Reviewer 3 Report

Comments and Suggestions for Authors

I understand that the authors of the ms have 

considered my comments and replied my concerns.

i am fine with revisions made. I don’t think the 

 need of other potential improvement. Therefore, 

I don’t have objection in regard to the acceptance 

of this work in it’s current form. 

Author Response

Comment1:I understand that the authors of the ms have 

considered my comments and replied my concerns.

i am fine with revisions made. I don’t think the 

 need of other potential improvement. Therefore, 

I don’t have objection in regard to the acceptance 

of this work in it’s current form. 

Response1: We sincerely appreciate the reviewer’s thorough evaluation and positive recommendation to accept the manuscript in its current form. The constructive feedback provided during the review process has significantly enhanced the clarity, rigor, and scientific impact of our work on NpO₂’s defect chemistry.

Reviewer 4 Report

Comments and Suggestions for Authors

In general, I am satisfied with authors responses and the changes that were made. The manuscript was improved by better comparison with literature data and better description of Np defects in anoxic and oxygen-rich atmosphere

I have just few remarks: 

  1. p. 6 line 206 - "fig.3" should be "Figure 3"
  2. p. 6 line 243 - Error - the quotation is missing 

Author Response

Comments 1: In general, I am satisfied with authors responses and the changes that were made. The manuscript was improved by better comparison with literature data and better description of Np defects in anoxic and oxygen-rich atmosphere

Response1We sincerely appreciate the reviewer’s positive assessment of the revised manuscript and their acknowledgment of the improved comparisons with literature data and defect environment analyses. We deeply value the reviewer’s constructive guidance, which has strengthened the manuscript’s scientific rigor and clarity. Should any further refinements be required, we remain fully committed to addressing them promptly

Comments 2: I have just few remarks: p. 6 line 206 - "fig.3" should be "Figure 3

Response 2: We sincerely appreciate the reviewer’s meticulous attention to detail in identifying the inconsistent figure citation format. This error has been corrected in the revised manuscript, and we have implemented additional safeguards to ensure full compliance with journal formatting standards.

As shown in Figure. 3, the relationship between Schottky defect formation energy and Fermi energy in NpO2 is systematically studied in this paper. [this change in the revised manuscript can be found in page 6, first paragraph, line 208-210]

Comments 3: p. 6 line 243 - Error - the quotation is missing 

Response 3: We sincerely appreciate the reviewer’s meticulous identification of the missing reference citation at Line 243. This oversight has been corrected in the revised manuscript, and we have implemented additional checks to ensure full consistency between in-text citations and the reference list.

We believe this contribution significantly advances the field’s understanding of defect-driven property modifications in nuclear materials. The viability of non-stoichiometric Schottky defects under anoxic conditions aligns with observations in isostructural CeO₂ and PuO2. [27,28] [this change in the revised manuscript can be found in page 6, last paragraph, line 242-245]
